# ENERGY-BASED CONCEPTUAL DIFFUSION MODEL

**Yi Qin[1], Xinyue Xu[1], Hao Wang[2†], Xiaomeng Li[1†]**
[1]The Hong Kong University of Science and Technology, [2]Rutgers University, [†]Equal advising
{yqinar, xxucb, eexmli}@ust.hk, hw488@cs.rutgers.edu

## ABSTRACT

Diffusion models have shown impressive sample generation capabilities across various domains. However, current methods are still lacking in human-understandable explanations and interpretable control: (1) they do not provide a probabilistic framework for systematic interpretation. For example, when tasked with generating an image of a "Nighthawk", they cannot quantify the probability of specific concepts (e.g., "black bill" and "brown crown" usually seen in Nighthawks) or verify whether the generated concepts align with the instruction. This limits explanations of the generative process; (2) they do not naturally support control mechanisms based on concept probabilities, such as correcting errors (e.g., correcting "black crown" to "brown crown" in a generated "Nighthawk" image) or performing imputations using these concepts, therefore falling short in interpretable editing capabilities. To address these limitations, we propose **Energy-based Conceptual Diffusion Models (ECDMs)**. ECDMs integrate diffusion models and Concept Bottleneck Models (CBMs) within the framework of Energy-Based Models to provide unified interpretations. Unlike conventional CBMs, which are typically discriminative, our approach extends CBMs to the generative process. ECDMs use a set of energy networks and pretrained diffusion models to define the joint energy estimation of the input instructions, concept vectors, and generated images. This unified framework enables concept-based generation, interpretation, debugging, intervention, and imputation through conditional probabilities derived from energy estimates. Our experiments on various real-world datasets demonstrate that ECDMs offer both strong generative performance and rich concept-based interpretability.

## 1 INTRODUCTION

Denoising diffusion probabilistic models are capable of generating high-quality images (Rombach et al., 2022; Bluethgen et al., 2024), videos (Brooks et al., 2024), and structured data (Ingraham et al., 2023) across various domains, such as artwork, medicine, and biology. However, existing diffusion models typically fall short in human-understandable explanations and interpretable control capabilities during the generation process. For instance, when the model is tasked with generating an image of a "Nighthawk", a practitioner may be interested in determining whether the model bases its generation on specific bird concepts (e.g., "black bill" and "brown crown" when generating a "Nighthawk" image). Additionally, the practitioner would want the capability to correct potential generation errors using these concepts (e.g., correcting "black crown" to "brown crown" in a generated "Nighthawk" image). Without these interpretation and correction capabilities, diffusion models – no matter how high-resolution their generated images are – can hardly be considered trustworthy or reliable by human standards.

Recent advances in interpretable diffusion models aim to address the problem by analyzing decomposed features (Du et al., 2021; 2023; Liu et al., 2022; 2023) or fine-tuning additional model components (Li et al., 2024a; Wang et al., 2023; Lyu et al., 2024; Luo et al., 2024; Li et al., 2024b; Kumari et al., 2023; Feng et al., 2022; Gandikota et al., 2023). However, these methods still suffer from the following key limitations:

1. **Systematic Interpretation:** They do not provide a probabilistic framework that facilitates systematic interpretation of the generation process. Consequently, it is still challenging to assess how the human-intended visual concepts are inherently represented and incorporated in the text-

to-image diffusion model's generation process, and whether the interpreted concepts from the generation process align with the intended concepts from the instruction.

2. **Concept-Based Generation:** They can only control the generation with a limited number of concepts (e.g., interpolating between "hairy" and "hairless" or composing a small number of visual components). As a result, they often struggle to generate images based on a broader set of concepts. This restriction significantly narrows the concept-based control space available in diffusion models, limiting their versatility in more complex generation tasks.

3. **Intervention:** Current methods often fail to correct generation errors based on concept-based probabilistic explanations (e.g., correcting "black crown" to "brown crown"). Furthermore, they cannot effectively intervene in the generation process by leveraging the interactions among class-level instructions, concept-based explanations, and sampling intermediates.

To provide systematic concept-based explanations and control for diffusion models, we propose **Energy-based Conceptual Diffusion Models** (ECDMs). ECDMs unify diffusion models and Concept Bottleneck Models (CBMs) under the Energy-Based Models framework. In contrast to conventional *discriminative* CBMs ("image" → "concepts" → "class label"), our ECDM enables concept-level interpretations and control to *generative* tasks ("class label" → "concepts" → "image").

Specifically, ECDMs use a set of networks and the pretrained diffusion model to quantify the energy between the class-level instruction $y$, concept-level explanation $c$, and the generated image $x$. Within this unified framework, one can

(1) generate the image $x$ with corresponding concept vectors $c$ as **interpretations**, i.e., $p(x, c|y)$,

(2) given an input instruction $y$ and the generated image $x$, **debug** what concepts are generated incorrectly by comparing the what concepts are generated (i.e., $p(c|x)$) and what concepts should have been generated (i.e., $p(c|y)$),

(3) given an input instruction $y$, **intervene** the generation process of image $x$ by replacing incorrect concepts with correct ones $[c_k]_{k=1}^{K-n}$, i.e., $p([c_k]_{k=K-n+1}^{K}, x|y, [c_k]_{k=1}^{K-n})$, and

(4) given an input instruction $y$ and part of a generated image $\Omega(x)$, **impute** the remainder of the image $\bar{\Omega}(x)$ with the concept explanations, i.e., $p(\bar{\Omega}(x), c|\Omega(x), y)$.

Importantly, thanks to the unified energy-based framework, these conditional probabilities can be naturally computed through composition of different energy functions. Our contributions are:

- We propose Energy-Based Conceptual Diffusion Models (ECDMs), a framework that unifies the concept-based generation, conditional interpretation, concept debugging, intervention, and imputation under the joint energy-based formulation.
- With ECDM's unified framework, we develop a set of algorithms to compute different conditional probabilities by composing corresponding energy functions.
- Empirical results on real-world datasets demonstrate ECDM's state-of-the-art performance in terms of image generation, imputation, and their conceptual interpretations.

## 2 ENERGY-BASED CONCEPTUAL DIFFUSION MODELS

In this section, we introduce the notation, problem settings, and then our proposed ECDM in detail.

**Notation.** We consider a class-level text-to-image generation setting, with $M$ classes and $K$ concepts. Specifically, given a class-level label $y$ (e.g., "Nighthawk"), a diffusion model will generate a corresponding image $x$, with the generation process potentially interpreted by a set of concepts, represented by a binary vector $c \in \mathcal{C} = \{0, 1\}^K$ (e.g., "black bill" and "brown crown"). We denote the $k$-th dimension of the concept vector $c$ as $c_k$. We denote the pretrained latent diffusion model as $\epsilon_\theta(\cdot, x_t, t)$, which is parameterized by $\theta$; it takes the noisy latent $x_t$ at timestep $t$ and the condition $\cdot$ as the input to predict the denoised latent $x_{t-1}$. We use a pretrained text encoder $F$ to extract (1) the class embedding $u$ from the given instruction ($u = F(y)$) and (2) the concept embedding $v$ from concepts ($v = F(c)$). Finally, the structured energy network $E_\psi(\cdot, \cdot)$ parameterized by $\psi$, maps $(x, c)$ or $(y, c)$ to real-valued scalar energy values.

**Problem Settings.** For each data point, we consider the following problem settings:

1. **Concept-Based Generation ($p(x, c|y)$).** This is the main task for a diffusion model. Given the instruction $y$, the goal is to infer the concepts $c$ and generate the image $x$. In ECDM, we decompose $p(x, c|y)$ into concept inference $p(c|y)$ and image generation $p(x|c)$.

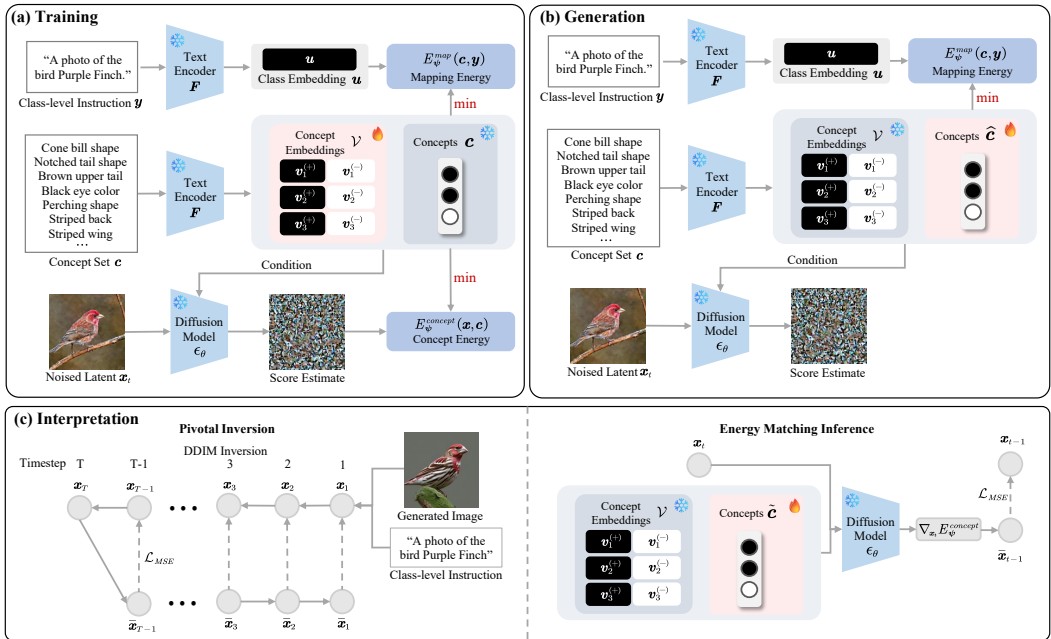

Figure 1: Overview of our ECDM. **(a) Training:** During training, the model learns the positive concept embedding $v_k^{(+)}$, the negative concept embedding $v_k^{(-)}$, and two sets of energy networks by optimizing Eqn. 1. **(b) Generation:** During generation, ECDMs first infer an optimal concept vector $\widehat{c}$, which is the most compatible with the instruction $y$, by minimizing the mapping energy, then use the inferred concept vector as the condition to minimize the concept energy by performing diffusion sampling. **(c) Interpretation:** During interpretation, ECDMs first inverse a pivotal trajectory using DDIM inversion given the generated image and corresponding instruction. Next, ECDMs update the concept probability $\widetilde{c}$ by minimizing the energy matching target (Eqn. 12).

2. **Interpretation $(p(c|x))$.** Interpret what concepts $c$ are used when generating the image $x$.

3. **Debugging $(p(c|y) \overset{?}{=} p(c|x))$.** Given the input $y$ and the generated image $x$, debug what concepts are generated *incorrectly* by comparing the what concepts are generated (i.e., $p(c|x)$) and what concepts should be generated (i.e., $p(c|y)$).

4. **Intervention/Correction $p([c_k]_{k=K-n+1}^K, x|y, [c_k]_{k=1}^{K-n})$.** Given the instruction $y$ and the *corrected* concepts $[c_k]_{k=1}^{K-n}$, infer other concepts $[c_k]_{k=K-n+1}^K$ and generate the image $x$.

5. **Imputation $p(\bar{\Omega}(x), c|\Omega(x), y)$.** Given the instruction $y$ and a partially masked image $\Omega(x)$, where $\Omega(\cdot)$ is a masking function and $x = \Omega(x) \cup \bar{\Omega}(x)$, impute the masked pixels $\bar{\Omega}(x)$ and generate the associated concept interpretations $c$.

## 2.1 Energy-Based Conceptual Diffusion Models

**Overview.** Our ECDM consists of two energy networks parameterized by $\psi$: (1) a concept energy network $E_\psi^{concept}(x, c)$, the gradient of which models the score of the concept-conditional data distribution $p(x|c)$ and has its minimum at the highest conditional log-likelihood and (2) a mapping energy network $E_\psi^{map}(y, c)$, which maps the class-level instruction $y$ to the corresponding concept vector $c$ by measuring the compatibility between $y$ and $c$. Both energy networks model the data distribution using "unnormalized" probability densities. Our ECDM is trained by minimizing the following loss function:

$$\mathcal{L}_{total}(x, c, y) = \mathcal{L}_{concept}(x, c) + \lambda_m \mathcal{L}_{map}(y, c), \quad (1)$$

where two terms $\mathcal{L}_{concept}$ and $\mathcal{L}_{map}$ denote the loss functions for the concept and mapping energy networks $E_\psi^{concept}(x, c)$ and $E_\psi^{map}(y, c)$, respectively. $\lambda_m$ is a balancing hyperparameter. Fig. 1 shows the overview of our ECDM. Below we provide rationale and details of the loss terms in detail.

**Generative Concept Energy Network $E_\psi^{concept}(x, c)$.** Our concept energy network captures the compatibility between the concepts $c$ and the generated image $x$ while enabling generative sam-

pling from the concept-conditional data distribution $p(\boldsymbol{x}|\boldsymbol{c})$. Notably, the gradient of the energy $E_{\boldsymbol{\psi}}^{concept}(\boldsymbol{x}, \boldsymbol{c})$ is proportional to the conditional data distribution $p_\theta(\boldsymbol{x}|\boldsymbol{c})$'s score, which is the diffusion model's denoising step $\epsilon_\theta(\boldsymbol{c}, \boldsymbol{x}, t)$. Formally we have:

$$\nabla_{\boldsymbol{x}} E_{\boldsymbol{\psi}}^{concept}(\boldsymbol{x}, \boldsymbol{c}) \propto \nabla_{\boldsymbol{x}} \log p_\theta(\boldsymbol{x}|\boldsymbol{c}) = \epsilon_\theta(\boldsymbol{c}, \boldsymbol{x}, t) \tag{2}$$

This enables the implicit modeling of this energy network using diffusion models. In practice, our concept energy network consists of an concept input network $D_c(\boldsymbol{c})$ and a pretrained diffusion network $\epsilon_\theta(\cdot, \boldsymbol{x}, t)$, where we replace $\boldsymbol{c}$ in $\epsilon_\theta(\boldsymbol{c}, \boldsymbol{x}, t)$ with $D_c(\boldsymbol{c})$. Specifically,

$$E_{\boldsymbol{\psi}}^{concept}(\boldsymbol{x}, \boldsymbol{c}) \triangleq \mathbb{E}_{\boldsymbol{x}, \epsilon \sim \mathcal{N}(\mathbf{0}, \boldsymbol{I}), t}[\|\epsilon - \epsilon_\theta(D_c(\boldsymbol{c}), \boldsymbol{x}_t, t)\|_2^2], \tag{3}$$

where the concept input network $D_c(\boldsymbol{c})$ works as follows: Given a set of $K$ concepts $\boldsymbol{c}$, each concept $k \in \{1, \ldots, K\}$ is associated with a positive embedding $\boldsymbol{v}_k^{(+)}$ and a negative embedding $\boldsymbol{v}_k^{(-)}$ projected by the text feature extractor $F$. The final concept embedding $\boldsymbol{v}_k$ is a combination of the positive and negative embedding weighted by the concept probability $c_k$, defined as $\boldsymbol{v}_k = c_k \cdot \boldsymbol{v}_k^{(+)} + (1 - c_k) \cdot \boldsymbol{v}_k^{(-)}$. Finally, another network $D_v(\boldsymbol{v})$ projects the combined concept embedding $\boldsymbol{v} \triangleq [\boldsymbol{v}_k]_{k=1}^K$ to the final input embedding, i.e., $D_c(\boldsymbol{c}) = D_v(\boldsymbol{v})$. Note that during training, we form the $\boldsymbol{v}_k$ as $\boldsymbol{v}_k^{(+)}$ if $c_k = 1$, and $\boldsymbol{v}_k^{(-)}$ if $c_k = 0$.

Since $E_{\boldsymbol{\psi}}^{concept}(\boldsymbol{x}, \boldsymbol{c})$ can be seen as the (approximate) variational upper bound for the negative log-likelihood $-\log p_\theta(\boldsymbol{x}|\boldsymbol{c})$ (more details in the Appendix D.2), it can be used directly as the loss function $\mathcal{L}_{concept}(\boldsymbol{x}, \boldsymbol{c})$ during training. We then have

$$\mathcal{L}_{concept}(\boldsymbol{x}, \boldsymbol{c}) \triangleq E_{\boldsymbol{\psi}}^{concept}(\boldsymbol{x}, \boldsymbol{c}) \triangleq \mathbb{E}_{\boldsymbol{x}, \epsilon \sim \mathcal{N}(\mathbf{0}, \boldsymbol{I}), t}[\|\epsilon - \epsilon_\theta(D_c(\boldsymbol{c}), \boldsymbol{x}_t, t)\|_2^2]. \tag{4}$$

After training, generating the image $\boldsymbol{x}$ given the concept vector $\boldsymbol{c}$ is then equivalent to solving $\boldsymbol{x} = \arg\min_{\boldsymbol{x}} E_{\boldsymbol{\psi}}^{concept}(\boldsymbol{x}, \boldsymbol{c})$ using Eqn. 2.

**Mapping Energy Network** $E_{\boldsymbol{\psi}}^{map}(\boldsymbol{y}, \boldsymbol{c})$**.** The mapping energy network connects the class-level instruction $\boldsymbol{y}$ and the concept vector $\boldsymbol{c}$ by measuring the compatibility between $\boldsymbol{y}$ and $\boldsymbol{c}$. We input the class embedding $\boldsymbol{u}$ corresponding to $\boldsymbol{y}$ and the fused concept embedding $\boldsymbol{w} = D_c(\boldsymbol{c})$ into a neural network to compute the mapping energy $E_{\boldsymbol{\psi}}^{map}(\boldsymbol{y}, \boldsymbol{c})$. Formally, we have:

$$E_{\boldsymbol{\psi}}^{map}(\boldsymbol{y}, \boldsymbol{c}) = D_{uw}(\boldsymbol{u}, \boldsymbol{w}), \tag{5}$$

where $D_{uw}(\cdot, \cdot)$ is a trainable neural network. The network will output an energy estimate for each pair of $(\boldsymbol{u}, \boldsymbol{w})$. Following (Xu et al., 2024), the training loss function for each instruction-concept pair $(\boldsymbol{y}, \boldsymbol{c})$ is formulated as:

$$\mathcal{L}_{map}(\boldsymbol{y}, \boldsymbol{c}) = E_{\boldsymbol{\psi}}^{map}(\boldsymbol{c}, \boldsymbol{y}) + \log\left(\sum_{m=1, \boldsymbol{c}' \in \mathcal{C}}^M e^{-E_{\boldsymbol{\psi}}^{map}(\boldsymbol{c}', \boldsymbol{y}_m)}\right), \tag{6}$$

where $\boldsymbol{c}'$ enumerates all concept combinations in the concept space $\mathcal{C}$. We use negative sampling to enumerate a subset of the possible combinations for computational efficiency.

## 2.2 CONCEPT-BASED JOINT GENERATION

Fig. 1(b) demonstrates the generation pipeline using our ECDM. To generate an image $\boldsymbol{x}$ based on concepts $\boldsymbol{c}$ given class-level instructions $\boldsymbol{y}$, we minimize the following joint energy:

$$E_{\boldsymbol{\psi}}^{joint}(\boldsymbol{x}, \boldsymbol{c}, \boldsymbol{y}) \triangleq E_{\boldsymbol{\psi}}^{concept}(\boldsymbol{x}, \boldsymbol{c}) + \lambda_m E_{\boldsymbol{\psi}}^{map}(\boldsymbol{c}, \boldsymbol{y}). \tag{7}$$

Specifically, concept-based generation aims to search for

$$\arg\max_{\widehat{\boldsymbol{x}}, \widehat{\boldsymbol{c}}} p(\widehat{\boldsymbol{x}}, \widehat{\boldsymbol{c}}|\boldsymbol{y}) = \arg\max_{\widehat{\boldsymbol{x}}, \widehat{\boldsymbol{c}}} \frac{e^{-E_{\boldsymbol{\psi}}^{joint}(\widehat{\boldsymbol{x}}, \widehat{\boldsymbol{c}}, \boldsymbol{y})}}{\sum_{\boldsymbol{x}, \boldsymbol{c}} e^{-E_{\boldsymbol{\psi}}^{joint}(\boldsymbol{x}, \boldsymbol{c}, \boldsymbol{y})}} = \arg\min_{\widehat{\boldsymbol{x}}, \widehat{\boldsymbol{c}}} E_{\boldsymbol{\psi}}^{joint}(\widehat{\boldsymbol{x}}, \widehat{\boldsymbol{c}}, \boldsymbol{y})$$

To make computation efficient, we start by searching for the optimal $\boldsymbol{c}$:

$$\arg\max_{\widehat{\boldsymbol{c}}} \ p(\widehat{\boldsymbol{c}}|\boldsymbol{y}) = \arg\min_{\widehat{\boldsymbol{c}}} \ E_{\boldsymbol{\psi}}^{map}(\boldsymbol{y}, \widehat{\boldsymbol{c}}). \tag{8}$$

After obtaining the optimal concept prediction $\widehat{c}$ which is the most compatible one with the instruction $y$, we use $\widehat{c}$ as the condition to minimize the joint energy model $E_{\psi}^{joint}(x, c, y)$ for generation. The minimization of the joint energy model is achieved by gradient descent-like sampling process from the diffusion model. Formally, we have:

$$x_{t-1} = x_t - \gamma \nabla_x E_{\psi}^{joint}(x, y, c)\big|_{x=x_t, c=\widehat{c}} + \xi, \tag{9}$$

$$= x_t - \gamma \nabla_x E_{\psi}^{concept}(x, c)\big|_{x=x_t, c=\widehat{c}} + \xi, \quad \xi \sim \mathcal{N}(0, \sigma_t^2 I), t = T, \dots, 1, \tag{10}$$

where $\nabla_x E_{\psi}^{concept}(x, c)$ is given by Eqn. 2. (See Appendix D.2 for more details.) We then alternate between Eqn. 8 and Eqn. 10 until convergence. Empirically, we find that one iteration usually produces sufficiently good results.

## 2.3 INTERPRETATION AND DEBUGGING VIA CONCEPT INVERSION

**Interpretation** $p(c|x)$. Our ECDM can interpret a given external diffusion model $\epsilon_{\phi}^{interpret}(y, x, t)$ using the conditional probability $p(c|x)$, which estimates what concepts $c$ are used by $\epsilon_{\phi}^{interpret}(y, x, t)$ to generate the image $x$ given the input instruction $y$. Specifically, we derive the concept probability by matching the energy landscape between our ECDM's concept energy network $E_{\psi}^{concept}(x, c)$ and the external energy model $E_{\theta}^{interpret}(x, y)$ associated with $\epsilon_{\phi}^{interpret}(y, x, t)$ (similar to Eqn. 2). Fig. 1(c) shows an overview of this process consisting of two steps: Pivotal Inversion and Energy Matching Inference.

**Pivotal Inversion.** Given an image $x$ and the corresponding instruction $y$, pivotal inversion aims to replay the sampling trajectory of the external (interpreted) energy model $E_{\theta}^{interpret}(x, y)$, providing pivotal representations at each sample step for alignment. We use the reversed DDIM (more details in Eqn. 39 of the Appendix) to produce a $T$-step deterministic trajectory between image $x_0$ and the Gaussian noise vector $x_T$. In each timestep $t$, the trajectory can be represented as:

$$\nabla_x E_{\phi}^{interpret}(x, y)\big|_{x=x_t} = \epsilon_{\phi}^{interpret}(y, x_t, t) \tag{11}$$

**Energy Matching Inference.** To infer the concept vector $c$ given the pivotal representation, we freeze the concept energy network $E_{\psi}^{concept}(x, c)$ to search for the optimal concept vector $\widetilde{c}$ globally at each timestep $t$ minimizing Eqn. 12 as follows:

$$\min \left\| \nabla_x E_{\psi}^{concept}(x, c) - \nabla_x E_{\theta}^{interpret}(x, y) \right\|_2^2, \tag{12}$$

Proposition 2.1 below shows that minimizing the Eqn. 12 is equivalent to matching the distribution between $p(c|x)$ and $p(y|x)$, thereby effectively finding the optimal concept vector $\widetilde{c}$ to interpret the external diffusion model's generation.

**Proposition 2.1** (**Conditional Concept Probability By Energy Matching**). *Given the instruction* $y$ *and the image* $x$, *minimizing Eqn. 12 is equivalent to minimizing the score's disparity between two conditional probabilities* $p(c|x)$ *and* $p(y|x)$:

$$\left\| \nabla_x E_{\psi}^{concept}(x, c) - \nabla_x E_{\theta}^{interpret}(x, y) \right\|_2^2 = \| \nabla_x \log p(c|x) - \nabla_x \log p(y|x) \|_2^2 \tag{13}$$

Transforming Proposition 2.1 into timestep-aware version, we can obtain the final optimal concept vector $\widetilde{c}$ via:

$$\arg \min_{\widetilde{c}} \left\| \nabla_x E_{\psi}^{concept}(x_t, \widetilde{c}) - \nabla_x E_{\theta}^{interpret}(x_t, y) \right\|_2^2 \tag{14}$$

**Debugging:** $p(c|y) \stackrel{?}{=} p(c|x)$. Debugging involves the comparison between what concepts the model has been generated ($p(c|x)$) and what concepts the model should have been generated ($p(y|x)$). $p(c|x)$ can be obtained via the energy matching process (Proposition 2.1), while $p(y|x)$ can be inferred by minimizing the mapping energy (Eqn. 8). By inspecting the disparity of these two conditional probabilities, users can pinpoint the potential cause of the generation error, laying the foundation for subsequent intervention and imputation to correct the discovered error.

## 2.4 Concept-Based Corrective Intervention and Imputation

**From Debugging to Intervention/Correction.** Based on the debugging results from Sec. 2.3, we can further perform concept intervention to correct the potential generation error. Specifically, if the debugging process in Sec. 2.3 finds that concepts $[c_k]_{k=1}^{K-n}$ are incorrect, i.e., $p([c_k]_{k=1}^{K-n}|\boldsymbol{y}) \neq p([c_k]_{k=1}^{K-n}|\boldsymbol{x})$, one can then intervene on the image generation process by correcting these concepts.

**Overview.** Specifically, ECDM's concept-based intervention consists of three steps: (1) correct concepts $[c_k]_{k=1}^{K-n}$ according to $p([c_k]_{k=1}^{K-n}|\boldsymbol{y})$, (2) given the corrected concepts, infer all remaining concepts via $p([c_k]_{k=K-n+1}^{K}|\boldsymbol{y}, [c_k]_{k=1}^{K-n})$, and (3) use all concepts to generate the image, i.e, computing $p(\boldsymbol{x}|[c_k]_{k=K-n+1}^{K}, \boldsymbol{y}, [c_k]_{k=1}^{K-n})$ via the concept energy network in Eqn. 3.

**Step 1: Correcting Concepts** $(p([c_k]_{k=1}^{K-n}|\boldsymbol{y}))$**.** Correcting concepts is straightforward. After computing the optimal $\widehat{c}$ by maximizing $p([c_k]_{k=1}^{K-n}|\boldsymbol{y})$ (Eqn. 8), one can simply set $c$ to $\widehat{c}$ in the ECDM.

**Step 2: Inferring Remaining Concepts.** Inference of the remaining concepts is facilitated by our mapping energy network and can be done using Eqn. 15 in Proposition 2.2 below.

**Proposition 2.2** (**Class-Specific Conditional Probability among Concepts**)**.** *Given partially concepts $[c_k]_{k=1}^{K-n}$ and class-level instruction $\boldsymbol{y}$, infer the remaining concepts $[c_k]_{k=K-n+1}^{K}$ is:*

$$p([c_k]_{k=K-n+1}^{K}|\boldsymbol{y}, [c_k]_{k=1}^{K-n}) = \frac{\frac{e^{-E_{\boldsymbol{\psi}}^{map}(\boldsymbol{c},\boldsymbol{y})}}{\sum_{\boldsymbol{c}'\in\mathcal{C}} e^{-E_{\boldsymbol{\psi}}^{map}(\boldsymbol{c}',\boldsymbol{y})}} \cdot p(\boldsymbol{y})}{\sum_{[c_j]_{j=K-n+1}^{K}} \frac{e^{-E_{\boldsymbol{\psi}}^{map}(\boldsymbol{c},\boldsymbol{y})}}{\sum_{\boldsymbol{c}'\in\mathcal{C}} e^{-E_{\boldsymbol{\psi}}^{map}(\boldsymbol{c}',\boldsymbol{y})}} \cdot p(\boldsymbol{y})} \tag{15}$$

**Step 3: Generating the Corrected Image.** Given all corrected concepts $\boldsymbol{c}$ ($[c_k]_{k=K-n+1}^{K}$ and $[c_k]_{k=1}^{K-n}$) combined), one then generates the corrected image $\boldsymbol{x}$ (i.e., $p(\boldsymbol{x}|\boldsymbol{c},\boldsymbol{y})$) using using Eqn. 10.

**Interpretable Concept-Based Imputation.** Additionally, ECDM can perform interpretable concept-based imputation based on the joint energy $E_{\boldsymbol{\psi}}^{joint}(\boldsymbol{x}, \boldsymbol{c}, \boldsymbol{y})$. We provide more details in Appendix B.

## 3 Experiments

### 3.1 Experiment Setup

**Datasets.** We use three real-world datasets to to evaluate different methods.

- **Animals with Attributes 2 (AWA2)** (Xian et al., 2018) is an animal image dataset containing 37,322 images, 85 concepts, and 50 animal classes. We select 45 photo-visible concepts for experiments, following ProbCBM (Kim et al., 2023). We only include animal classes that contain more than 300 images, leading to a total number of 24 classes in our final dataset.
- **Caltech-UCSD Birds-200-2011 (CUB)** (Wah et al., 2011) is a fine-grained bird image dataset with 11,788 images, 312 annotated attributes, and 200 classes. Following previous works (Koh et al., 2020; Kim et al., 2023; Zarlenga et al., 2022), we select 112 attributes as the 112 concepts.
- **CelebA-HQ** (Karras, 2017) is a high-quality face image dataset with 30,000 images, 40 binary attributes and 10,177 identities. Following CEM (Zarlenga et al., 2022), we select 8 most frequent attributes as the 8 concepts and use 6 combination of the selected attributes as the 6 classes in our setting.

**Baseline and Implementation Details.** We compare the generation results of ECDM with the direct class-level instruction generation of Stable Diffusion 2.1 (**SD-2.1**) (Rombach et al., 2022) and **PixArt-**$\alpha$ (Chen et al., 2023). We further include the generation result from Text Inversion (**TI**) (Gal et al., 2022), which is the most related finetuning-based method. We build our model upon the pretrained Stable Diffusion 2.1 (Rombach et al., 2022) with parameters frozen for all experiments. We use the AdamW optimizer during the training and inference process.

**Evaluation Metrics.** We employ three specific metrics to evaluate different methods:

Table 1: The generation quality evaluation results on different datasets. Textual Inversion is not readily available in PixArt-$\alpha$ model, therefore unavailable for the experiment. The Textual Inversion results of CelebA-HQ is based on SD-2.1, hence identical results, see Appendix E for further explanation.

| Data Model | CUB | | | AWA2 | | | CelebA-HQ | | |
|---|---|---|---|---|---|---|---|---|---|
| Metric | FID | Class Accuracy | Concept Accuracy | FID | Class Accuracy | Concept Accuracy | FID | Class Accuracy | Concept Accuracy |
| SD-2.1 | 29.55 | 0.5033 | 0.9222 | 37.79 | 0.8935 | 0.9850 | 53.47 | 0.4881 | 0.8079 |
| PixArt-$\alpha$ | 46.85 | 0.1208 | 0.8231 | 59.71 | 0.9008 | 0.9764 | - | - | - |
| TI | 23.36 | 0.6397 | 0.9496 | 29.63 | 0.9142 | **0.9863** | 53.47 | 0.4881 | 0.8079 |
| **ECDM (Ours)** | **22.94** | **0.6492** | **0.9561** | **28.91** | **0.9200** | 0.9801 | **52.89** | **0.5017** | **0.8182** |

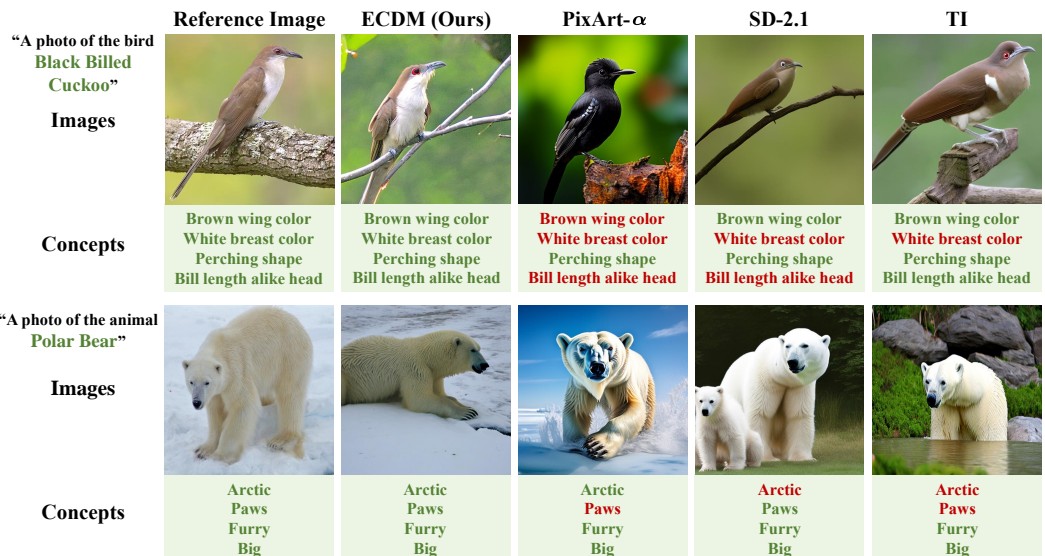

Figure 2: Visualizing generated outputs on CUB (upper) and AWA2 (lower) datasets. Words in green/red indicate a correctly/wrongly generated visual concept. Images are generated under the same random seed and instruction. Our ECDM generates more fine-grained and correct details compared to other methods (e.g., "white breast color" and "bill length alike head" in Row 1).

- **Frechet Inception Distance (FID).** We measure the FID (Heusel et al., 2017) between the synthetic and real images to evaluate the generated image quality. Lower FID indicates higher image generation quality.
- **Class Accuracy.** We train three class-level ResNet101 classification models (He et al., 2016) on the corresponding datasets, and use the trained model to measure the class accuracy of generated images. Higher class accuracy suggests that the generated images more effectively capture the defining characteristics of a class.
- **Concept Accuracy.** We calculate the concept accuracy between the ground-truth concepts and the predicted concepts from pretrained CEMs (Zarlenga et al., 2022). Higher concept accuracy indicates that the generated image covers more desired visual concepts.

See more details on dataset construction, implementations, and evaluation in Appendix E and F.

## 3.2 RESULTS

**Concept-Based Joint Generation.** Fig. 2 shows the generation results of our ECDM on different datasets. Visually, the outputs of our model are better aligned with the characteristics of real-world subjects and exhibit more refined details compared to both standard text-to-image diffusion models and their fine-tuned variants. The visual concepts included in the reference (ground-truth) image's (marked in green) are comprehensively depicted in our ECDM's generated images. For instance,

| Class Name | | Concept Names | Grey breast color | Yellow belly color | Round wing shape | Brown upper color | All-purpose bill shape | Black eye color |
|---|---|---|---|---|---|---|---|---|
| Great Crested Flycatcher | | Was Generated $p(c|x)$ | 0.0235 | 0.0488 | 0.3806 | 0.9363 | 0.9661 | 0.9884 |
| | | Should Generate $p(c|y)$ | 0.9998 | 1.0000 | 0.9981 | 0.9910 | 0.9947 | 1.0000 |
| | | Ground Truth | 1 | 1 | 1 | 1 | 1 | 1 |
| | | Concept Names | Brown wing color | Grey wing color | Solid tail pattern | Perching shape | Grey crown color | Blue belly color |
| Olive Sided Flycatcher | | Was Generated $p(c|x)$ | 0.8961 | 0.3984 | 0.0684 | 0.9866 | 0.8724 | 0.0721 |
| | | Should Generate $p(c|y)$ | 0.0021 | 0.9975 | 1.0000 | 0.9991 | 0.9950 | 0.0074 |
| | | Ground Truth | 0 | 1 | 1 | 1 | 1 | 0 |

Figure 3: Interpretation results on the CUB dataset. The images $x$ are generated from an *external pretrained diffusion model* (i.e., vanilla SD-2.1). Numbers in red indicate potential generation errors compared with real concepts. Our ECDM can correctly interpret what concepts were generated ($p(c|x)$) and what concepts should be generated for instruction $y$ ($p(c|y)$).
the concepts "white breast color" and "bill length alike head" of the "Black Billed Cuckoo" are successfully generated in the image. In contrast, all other methods miss the concept "white breast color", and both PixArt-$\alpha$ and SD-2.1 miss the concept "bill length alike head".

Table 1 shows the quantitative results. Our ECDM consistently achieves a lower FID compared to the baselines, indicating that ECDM produces images with higher fidelity and quality. Notably, the class and concept accuracy of our model's generated images in the majority of datasets outperforms all other methods. This suggests that our model incorporates more visible concepts during generation, providing richer class-discriminative characteristics in the resulting images.

**Interpretation via Concept Inversion.** Fig. 3 shows our ECDM's probabilistic interpretations of the generation process based on visual concepts. It shows that ECDM's inferred concept probabilities (the row "Was Generated $p(c|x)$) correctly reflect the concepts generated by the model. Additionally, the concept probabilities derived from the mapping energy network (the row "Should Generate $p(c|y)$") correctly reflect the concepts that should be generated for the specific class (e.g., "Great Crested Flycatcher").

**Debugging by Comparing $p(c|x)$ and $p(c|y)$.** By comparing what concepts were generated ($p(c|x)$) and what concepts should be generated for class $y$ ($p(c|y)$), we can identify the cause of potential generation errors. For example, an external pretrained diffusion model generates an "Olive Sided Flycatcher" with "brown wings", although it should be "grey wings". Our ECDM assigns the concept "brown wing color" a high prediction probability (0.8961), suggesting it was a key factor in the generation. Our ECDM's further indicates that "brown wing color" should *not* be generated, with the "Should Generate" probability $p(c|y) = 0.0021$. In this way, users can identify incorrectly predicted concept probabilities using our method, gaining insight into the model's generative tendencies and establishing a foundation for further interpretive interventions and corrections.

**Concept-Based Intervention.** Fig. 5 shows the intervention results based on interpreted concept probabilities. After user intervention, ECDM can effectively correct generation errors related to visual concepts. For example, the interpretation process revealed that the "Black Billed Cuckoo" should not have been generated with the concepts "grey crown color" and "grey upper color", but rather with "white breast color" and "perching shape." After the user intervened by providing the correct concept set, the model successfully corrected the generation based on these proper concepts.

## 4 CONCLUSION AND LIMITATIONS

In this paper, we extend the concept bottleneck model into the generative process, identifying the need for a joint modeling of conceptual generation, interpretation, debugging, intervention, and imputation. We proposed Energy-Based Conceptual Diffusion Model (ECDM), a framework that unifies generation, conditional interpretation and debugging, sampling intervention and imputation under the joint energy-based formulation. A set of conditional probabilities is derived through the combination of the energy functions. Our work also has several limitations, including the need for more precise regional control in concept-based editing and the requirement for concept ground truth.

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

# A    RELATED WORKS AND PRELIMINARIES

## A.1    RELATED WORKS

**Energy-Based Modeling of Diffusion Models** convert diffusion models into energy-based models (EBMs) (Salimans & Ho, 2021). In (Liu et al., 2022), the generation process of the diffusion model can be decomposed into a linear combination of individual factors (Du et al., 2021), each represented by a different EBM. COMET and its extension (Du et al., 2021; Su et al., 2024) trained energy functions by recomposing input images to discover global concepts and scene objects. Furthermore, Liu et al. (2023) integrated EBM-based concept discovery and compositional processes into text-to-image diffusion models, while Du et al. (2023) improved the sampling strategy and proposed a new parameterization scheme for compositional operators and samplers in energy-based diffusion models. We note several key differences between these methods and our ECDM. (1) The number of supported concepts is fixed and limited (e.g., only 6 concepts (Su et al., 2024), compared to 112 concepts in our ECDM), and hence not sufficiently informative as interpretations. (2) More importantly, these works aim to compositional generation with deterministic concepts, therefore fail to provide probabilistic interpretation, which is the focus of our ECDM. Therefore these methods are *not applicable for our setting*.

In contrast, our ECDMs explicitly consider human-understandable probabilistic concept explanations in its design by jointly modeling the input instruction $\boldsymbol{y}$, associated concepts $\boldsymbol{c}$, and the generated image $\boldsymbol{x}$ during the generation process within a unified energy-based framework.

**Concept Bottleneck Models** (CBMs) (Kumar et al., 2009; Koh et al., 2020) first predict a set of human-understandable concepts given an input, and then use the predicted concept vector to infer the final model decisions. Built upon the original CBMs, Concept Embedding Models (CEMs) (Zarlenga et al., 2022) encode each concept into a positive and a negative embedding, which are activated accordingly based on the presence or absence of the corresponding concept. Energy-based Concept Bottleneck Models (ECBMs) (Xu et al., 2024) formulate the CBMs under the EBM framework, successfully improving both concept and class-label accuracy. However, these CBMs are *discriminative*, focusing on predicting concepts and labels given an image; they cannot generate images from labels or concepts and are therefore *not applicable to our setting*.

**Interpretable Diffusion Models** employ adaptors (Gandikota et al., 2023; Lyu et al., 2024) or additional learning procedures (Wang et al., 2023; Guo et al., 2023; Ismail et al., 2023; Luo et al., 2024; Hudson et al., 2024) to discover interpretable generation directions towards certain concepts (e.g., face attributes) or objects. Among them, most related to our work are EGC (Guo et al., 2023) and CBGM (Ismail et al., 2023). EGC (Guo et al., 2023) learns a diffusion model to perform both generation and classification via energy-based formulation, while CBGM (Ismail et al., 2023) integrates a concept bottleneck in the diffusion model to enhance its interpretability. However, both methods require training a new diffusion model from scratch and are therefore *not applicable to our setting*, which focuses on explaining and finetuning pretrained large diffusion models.

## A.2    PRELIMINARIES ON CONDITIONAL DIFFUSION MODELS

Conditional diffusion models aim to learn a data distribution $p(\boldsymbol{x}|\boldsymbol{y})$ by gradually removing noise from a normally distributed variable. This process is equivalent to learning the reverse trajectory of a fixed Markov chain of length $T$. These models can also be interpreted as a sequence of denoising networks $\epsilon_\theta(\boldsymbol{y}, \boldsymbol{x}_t, t)$, where $t = 1, \ldots, T$. Each autoencoder is trained to predict a noise-free variant of its noisy input $\boldsymbol{x}_t$. The corresponding objective can be simplified as follows:

$$L_{CDM} = \mathbb{E}_{\boldsymbol{x}, \epsilon \sim \mathcal{N}(\boldsymbol{0}, \boldsymbol{I}), t}[\|\epsilon - \epsilon_\theta(\boldsymbol{y}, \boldsymbol{x}_t, t)\|_2^2], \tag{16}$$

where $t$ is uniformly sampled from $\{1, \ldots, T\}$. Ho et al. (2020) show that minimizing Eqn. 16 is equivalent to minimizing the variational bound on negative log likelihood of the data distribution:

$$\mathbb{E}[-\log p_\theta(\boldsymbol{x}|\boldsymbol{y})] \leq \mathbb{E}_{\boldsymbol{x}, \epsilon \sim \mathcal{N}(\boldsymbol{0}, \boldsymbol{I}), t}[\|\epsilon - \epsilon_\theta(\boldsymbol{y}, \boldsymbol{x}_t, t)\|_2^2] := \mathcal{L}_{CDM} \tag{17}$$

After training, the diffusion model generates an image $\boldsymbol{x}_0$ by iterative denoising, starting from initial noise $\boldsymbol{x}_T \sim \mathcal{N}(\boldsymbol{0}, \boldsymbol{I})$ and continuing the sampling steps as follows:

$$\boldsymbol{x}_{t-1} = \boldsymbol{x}_t - \gamma \cdot \epsilon_\theta(\boldsymbol{y}, \boldsymbol{x}_t, t) + \eta \cdot \xi, \quad \xi \sim \mathcal{N}(\boldsymbol{0}, \sigma_t^2 \boldsymbol{I}), \tag{18}$$

where $\gamma$ is the step size, and $\eta$ is the randomness-controlling parameter in DDIM (Song et al., 2020a). Song et al. (2020b) further show that the diffusion model trained by Eqn. 16 also models the score of the given data distribution, i.e., $\epsilon_\theta(\boldsymbol{y}, \boldsymbol{x}_t, t) = \nabla_{\boldsymbol{x}} \log p_\theta(\boldsymbol{x}|\boldsymbol{y})|_{\boldsymbol{x}=\boldsymbol{x}_t}$. Note that one can replace the input instruction $\boldsymbol{y}$ with a concept vector $\boldsymbol{c}$ to learn $p(\boldsymbol{x}|\boldsymbol{c})$ by training $\epsilon_\theta(\boldsymbol{c}, \boldsymbol{x}_t, t)$.

## B  DETAILS ON INTERPRETABLE CONCEPT-BASED IMPUTATION

**Imputation by Sampling from** $p(\bar{\Omega}(\boldsymbol{x}), \boldsymbol{c}|\Omega(\boldsymbol{x}), \boldsymbol{y})$. Our ECDM can also perform image imputation with concept-based interpretations. Specifically, given the input instruction $\boldsymbol{y}$ and the partial image $\Omega(\boldsymbol{x})$, it can generate (impute) the remaining pixels of the image $\bar{\Omega}(\boldsymbol{x})$ and the associated concepts $\boldsymbol{c}$ as concept-based interpretations. This is done via Eqn. 19 in Proposition B.1 below.

**Proposition B.1** (**Conditional Sampling by Concept Explaination**). *Given partially image $\Omega(\boldsymbol{x})$ and class-level instruction $\boldsymbol{y}$, inferring the remainder of the image $\bar{\Omega}(\boldsymbol{x})$ and concepts $\boldsymbol{c}$ corresponds to computing:*

$$p(\bar{\Omega}(\boldsymbol{x}), \boldsymbol{c}|\Omega(\boldsymbol{x}), \boldsymbol{y}) \propto \frac{e^{-E_\psi^{joint}(\boldsymbol{x},\boldsymbol{c},\boldsymbol{y})}}{\sum_{\boldsymbol{x}} e^{-E_\psi^{joint}(\boldsymbol{x},\boldsymbol{c},\boldsymbol{y})}} \cdot \frac{e^{-E_\psi^{map}(\boldsymbol{c},\boldsymbol{y})}}{\sum_{\boldsymbol{c}' \in \mathcal{C}} e^{-E_\psi^{map}(\boldsymbol{c}',\boldsymbol{y})}} \cdot p(\boldsymbol{y}) \qquad (19)$$

The proof is available in Appendix D.1. Specifically, one can obtain the imputed image part $\bar{\Omega}(\boldsymbol{x})$ and the concept-based interpretations $\boldsymbol{c}$ by solving $\arg\max_{\bar{\Omega}(\boldsymbol{x}),\boldsymbol{c}} p(\bar{\Omega}(\boldsymbol{x}), \boldsymbol{c}|\Omega(\boldsymbol{x}), \boldsymbol{y})$ above.

**Results for Interpretable Imputation.** Fig. 4 further demonstrates the imputation results from our model and the standard SD-2.1-Inpainting model. Compared to the standard inpainting model, ECDM better preserves class-specific characteristics (e.g., the bill of the Vermilion Flycatcher should be black, and the forehead should not be grey) based on the inferred concepts. Our model also consistently emphasizes the visual concepts related to the area being imputed (e.g., more white breast and throat areas in the imputed region of the Black Billed Cuckoo). These two examples demonstrate that ECDM effectively harnesses both concept perception and concept-based generation capabilities.

## C  ADDITIONAL RESULT VISUALIZATION

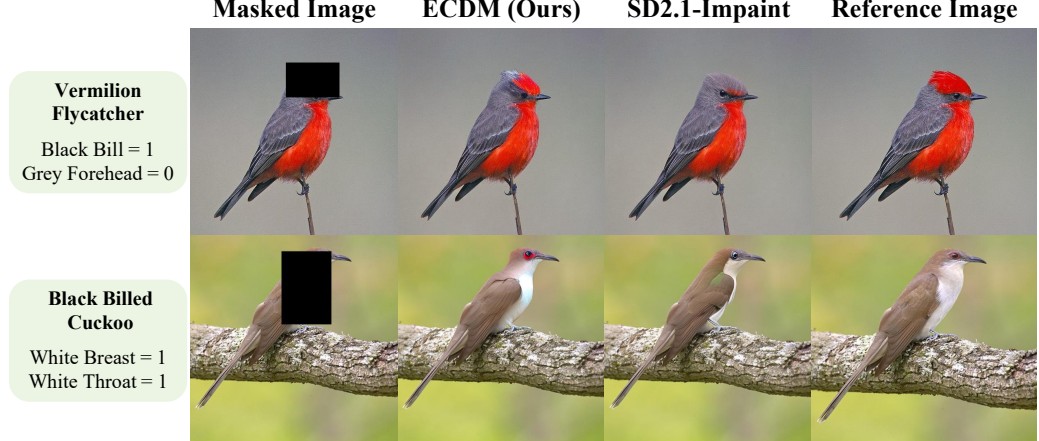

Figure 4: Imputation on the CUB dataset. The imputation results of our ECDM is more consistent with the corresponding concepts (e.g., "Grey Forehead = 0" in Row 1).

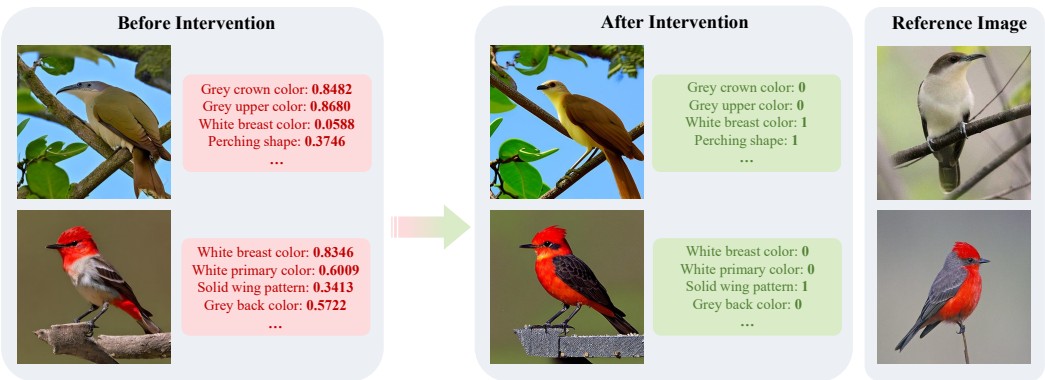

Figure 5: Intervention visualization on CUB dataset. Contents in red are concepts debugged by ECDM. Concept sets are corrected to intervene the generation process (e.g., the "White breast color" in the Row 2 image is effectively intervened and corrected to red color).

## D    PROOFS AND ADDITIONAL DISCUSSIONS

### D.1    PROOFS

**Proposition 2.1** (**Conditional Concept Probability By Energy Matching**). *Given the instruction* $\boldsymbol{y}$ *and the image* $\boldsymbol{x}$, *minimizing Eqn.* 12 *is equivalent to minimizing the score's disparity between two conditional probabilities* $p(\boldsymbol{c}|\boldsymbol{x})$ *and* $p(\boldsymbol{y}|\boldsymbol{x})$:

$$\left\| \nabla_{\boldsymbol{x}} E_{\boldsymbol{\psi}}^{concept}(\boldsymbol{x}, \boldsymbol{c}) - \nabla_{\boldsymbol{x}} E_{\theta}^{interpret}(\boldsymbol{x}, \boldsymbol{y}) \right\|_2^2 = \left\| \nabla_{\boldsymbol{x}} \log p(\boldsymbol{c}|\boldsymbol{x}) - \nabla_{\boldsymbol{x}} \log p(\boldsymbol{y}|\boldsymbol{x}) \right\|_2^2 \qquad (13)$$

*Proof.* For $p(\boldsymbol{x}|\boldsymbol{c})$ we have:

$$p(\boldsymbol{x}|\boldsymbol{c}) = \frac{p(\boldsymbol{c}|\boldsymbol{x}) \cdot p(\boldsymbol{x})}{p(\boldsymbol{c})}. \qquad (20)$$

Therefore,

$$\begin{aligned} \nabla_{\boldsymbol{x}} \log p(\boldsymbol{x}|\boldsymbol{c}) &= \nabla_{\boldsymbol{x}} \log \frac{p(\boldsymbol{c}|\boldsymbol{x}) \cdot p(\boldsymbol{x})}{p(\boldsymbol{c})} \\ &= \nabla_{\boldsymbol{x}} \log p(\boldsymbol{c}|\boldsymbol{x}) + \nabla_{\boldsymbol{x}} \log p(\boldsymbol{x}). \end{aligned} \qquad (21)$$

For $p(\boldsymbol{x}|\boldsymbol{y})$ we have:

$$p(\boldsymbol{x}|\boldsymbol{y}) = \frac{p(\boldsymbol{y}|\boldsymbol{x}) \cdot p(\boldsymbol{x})}{p(\boldsymbol{y})}. \qquad (22)$$

Therefore, by a similar argument,

$$\begin{aligned} \nabla_{\boldsymbol{x}} \log p(\boldsymbol{x}|\boldsymbol{y}) &= \nabla_{\boldsymbol{x}} \log \frac{p(\boldsymbol{y}|\boldsymbol{x}) \cdot p(\boldsymbol{x})}{p(\boldsymbol{y})} \\ &= \nabla_{\boldsymbol{x}} \log p(\boldsymbol{y}|\boldsymbol{x}) + \nabla_{\boldsymbol{x}} \log p(\boldsymbol{x}). \end{aligned} \qquad (23)$$

Given Eqn. 21 and Eqn. 23, we have:

$$\begin{aligned} \left\| \nabla_{\boldsymbol{x}} \log p(\boldsymbol{x}|\boldsymbol{c}) - \nabla_{\boldsymbol{x}} \log p(\boldsymbol{x}|\boldsymbol{y}) \right\|_2^2 &= \left\| \nabla_{\boldsymbol{x}} \log p(\boldsymbol{c}|\boldsymbol{x}) - \nabla_{\boldsymbol{x}} \log p(\boldsymbol{y}|\boldsymbol{x}) \right\|_2^2 \\ &= \left\| \nabla_{\boldsymbol{x}} E_{\boldsymbol{\psi}}^{concept}(\boldsymbol{x}, \boldsymbol{c}) - \nabla_{\boldsymbol{x}} E_{\theta}^{interpret}(\boldsymbol{x}, \boldsymbol{y}) \right\|_2^2, \end{aligned} \qquad (24)$$

concluding the proof.    □

**Proposition 2.2** (**Class-Specific Conditional Probability among Concepts**). *Given partially concepts* $[\boldsymbol{c}_k]_{k=1}^{K-n}$ *and class-level instruction* $\boldsymbol{y}$, *infer the remaining concepts* $[\boldsymbol{c}_k]_{k=K-n+1}^{K}$ *is:*

$$p([\boldsymbol{c}_k]_{k=K-n+1}^{K}|\boldsymbol{y},[\boldsymbol{c}_k]_{k=1}^{K-n}) = \frac{\frac{e^{-E_{\psi}^{map}(\boldsymbol{c},\boldsymbol{y})}}{\sum_{\boldsymbol{c}'\in\mathcal{C}} e^{-E_{\psi}^{map}(\boldsymbol{c}',\boldsymbol{y})}} \cdot p(\boldsymbol{y})}{\sum_{[\boldsymbol{c}_j]_{j=K-n+1}^{K}} \frac{e^{-E_{\psi}^{map}(\boldsymbol{c},\boldsymbol{y})}}{\sum_{\boldsymbol{c}'\in\mathcal{C}} e^{-E_{\psi}^{map}(\boldsymbol{c}',\boldsymbol{y})}} \cdot p(\boldsymbol{y})} \tag{15}$$

*Proof.* We denote the mapping energy of the energy network parameterized by $\psi$ between concept $\boldsymbol{c}$ and the label $\boldsymbol{y}$ as $E_{\psi}^{map}(\boldsymbol{c},\boldsymbol{y})$. We have:

$$p(\boldsymbol{c}|\boldsymbol{y}) = \frac{e^{-E_{\psi}^{map}(\boldsymbol{c},\boldsymbol{y})}}{\sum_{\boldsymbol{c}'\in\mathcal{C}} e^{-E_{\psi}^{map}(\boldsymbol{c}',\boldsymbol{y})}}. \tag{25}$$

By Bayes rule, we then have:

$$
\begin{aligned}
p([\boldsymbol{c}_k]_{k=K-n+1}^{K}|\boldsymbol{y},[\boldsymbol{c}_k]_{k=1}^{K-n}) &= \frac{p([\boldsymbol{c}_k]_{k=K-n+1}^{K},[\boldsymbol{c}_k]_{k=1}^{K-n},\boldsymbol{y})}{p([\boldsymbol{c}_k]_{k=1}^{K-n},\boldsymbol{y})} \\
&= \frac{p(\boldsymbol{c},\boldsymbol{y})}{p([\boldsymbol{c}_k]_{k=1}^{K-n},\boldsymbol{y})} \\
&= \frac{p(\boldsymbol{c}|\boldsymbol{y})\cdot p(\boldsymbol{y})}{p([\boldsymbol{c}_k]_{k=1}^{K-n},\boldsymbol{y})} \\
&= \frac{p(\boldsymbol{c}|\boldsymbol{y})\cdot p(\boldsymbol{y})}{\sum_{[\boldsymbol{c}_j]_{j=K-n+1}^{K}} p(\boldsymbol{c}|\boldsymbol{y})\cdot p(\boldsymbol{y})} \\
&= \frac{\frac{e^{-E_{\psi}^{map}(\boldsymbol{c},\boldsymbol{y})}}{\sum_{\boldsymbol{c}'\in\mathcal{C}} e^{-E_{\psi}^{map}(\boldsymbol{c}',\boldsymbol{y})}}\cdot p(\boldsymbol{y})}{\sum_{[\boldsymbol{c}_j]_{j=K-n+1}^{K}} \frac{e^{-E_{\psi}^{map}(\boldsymbol{c},\boldsymbol{y})}}{\sum_{\boldsymbol{c}'\in\mathcal{C}} e^{-E_{\psi}^{map}(\boldsymbol{c}',\boldsymbol{y})}}\cdot p(\boldsymbol{y})},
\end{aligned}
\tag{26}
$$

concluding the proof. $\square$

**Proposition B.1** (**Conditional Sampling by Concept Explaination**). *Given partially image* $\Omega(\boldsymbol{x})$ *and class-level instruction* $\boldsymbol{y}$, *inferring the remainder of the image* $\bar{\Omega}(\boldsymbol{x})$ *and concepts* $\boldsymbol{c}$ *corresponds to computing:*

$$p(\bar{\Omega}(\boldsymbol{x}),\boldsymbol{c}|\Omega(\boldsymbol{x}),\boldsymbol{y}) \propto \frac{e^{-E_{\psi}^{joint}(\boldsymbol{x},\boldsymbol{c},\boldsymbol{y})}}{\sum_{\boldsymbol{x}} e^{-E_{\psi}^{joint}(\boldsymbol{x},\boldsymbol{c},\boldsymbol{y})}} \cdot \frac{e^{-E_{\psi}^{map}(\boldsymbol{c},\boldsymbol{y})}}{\sum_{\boldsymbol{c}'\in\mathcal{C}} e^{-E_{\psi}^{map}(\boldsymbol{c}',\boldsymbol{y})}} \cdot p(\boldsymbol{y}) \tag{19}$$

*Proof.* Given Eqn. 35 and Eqn. 25, we have:

$$
\begin{aligned}
p(\boldsymbol{x},\boldsymbol{c},\boldsymbol{y}) &= p(\boldsymbol{x}|\boldsymbol{c},\boldsymbol{y})\cdot p(\boldsymbol{c},\boldsymbol{y}) \\
&= p(\boldsymbol{x}|\boldsymbol{c},\boldsymbol{y})\cdot p(\boldsymbol{c}|\boldsymbol{y})\cdot p(\boldsymbol{y}) \\
&= \frac{e^{-E_{\psi}^{joint}(\boldsymbol{x},\boldsymbol{c},\boldsymbol{y})}}{\sum_{\boldsymbol{x}} e^{-E_{\psi}^{joint}(\boldsymbol{x},\boldsymbol{c},\boldsymbol{y})}} \cdot \frac{e^{-E_{\psi}^{map}(\boldsymbol{c},\boldsymbol{y})}}{\sum_{\boldsymbol{c}'\in\mathcal{C}} e^{-E_{\psi}^{map}(\boldsymbol{c}',\boldsymbol{y})}} \cdot p(\boldsymbol{y}).
\end{aligned}
\tag{27}
$$

We already have $\boldsymbol{x} = \Omega(\boldsymbol{x}) \cup \bar{\Omega}(\boldsymbol{x})$, and given Eqn. 27 we can get:

$$
\begin{aligned}
p(\bar{\Omega}(\boldsymbol{x}), \boldsymbol{c}|\Omega(\boldsymbol{x}), \boldsymbol{y}) &= \frac{p(\Omega(\boldsymbol{x}), \bar{\Omega}(\boldsymbol{x}), \boldsymbol{c}|\boldsymbol{y})}{p(\Omega(\boldsymbol{x})|\boldsymbol{y})} \\
&= \frac{p(\boldsymbol{x}, \boldsymbol{c}|\boldsymbol{y})}{p(\Omega(\boldsymbol{x})|\boldsymbol{y})} \\
&= \frac{p(\boldsymbol{x}, \boldsymbol{c}|\boldsymbol{y})}{\sum\limits_{\bar{\Omega}(\boldsymbol{x})} p(\Omega(\boldsymbol{x}), \bar{\Omega}(\boldsymbol{x})|\boldsymbol{y})} \\
&= \frac{p(\boldsymbol{x}, \boldsymbol{c}|\boldsymbol{y})}{\sum\limits_{\bar{\Omega}(\boldsymbol{x})} p(\boldsymbol{x}|\boldsymbol{y})} \\
&= \frac{\dfrac{p(\boldsymbol{x}, \boldsymbol{c}, \boldsymbol{y})}{p(\boldsymbol{y})}}{\sum\limits_{\bar{\Omega}(\boldsymbol{x})} p(\boldsymbol{x}|\boldsymbol{y})} \\
&= \frac{p(\boldsymbol{x}, \boldsymbol{c}, \boldsymbol{y})}{\sum\limits_{\bar{\Omega}(\boldsymbol{x})} p(\boldsymbol{x}|\boldsymbol{y}) \cdot p(\boldsymbol{y})} \\
&\propto p(\boldsymbol{x}, \boldsymbol{c}, \boldsymbol{y}) \\
&\propto \frac{e^{-E_{\psi}^{joint}(\boldsymbol{x}, \boldsymbol{c}, \boldsymbol{y})}}{\sum_{\boldsymbol{x}} e^{-E_{\psi}^{joint}(\boldsymbol{x}, \boldsymbol{c}, \boldsymbol{y})}} \cdot \frac{e^{-E_{\psi}^{map}(\boldsymbol{c}, \boldsymbol{y})}}{\sum_{\boldsymbol{c'} \in \mathcal{C}} e^{-E_{\psi}^{map}(\boldsymbol{c'}, \boldsymbol{y})}} \cdot p(\boldsymbol{y}),
\end{aligned}
\tag{28}
$$

concluding the proof. $\qquad\square$

## D.2   ADDITIONAL DISCUSSION ON CONCEPT ENERGY NETWORK

We provide more details on the association between the concept energy network $E_{\psi}^{concept}(\boldsymbol{x}, \boldsymbol{c})$ and the negative log-likelihood of the conditional data distribution $-\log p_\theta(\boldsymbol{x}|\boldsymbol{c})$. According to Ho et al. (2020), optimizing the variational bound for the conditional data distribution's negative log likelihood in diffusion model has:

$$
\mathbb{E}[-\log p_\theta(\boldsymbol{x}_0|\boldsymbol{c})] \le \mathbb{E}_{q(\boldsymbol{x}_{0:T})}[-\log \frac{p_\theta(\boldsymbol{x}_{0:T}|\boldsymbol{c})}{q(\boldsymbol{x}_{1:T}|\boldsymbol{x}_0, \boldsymbol{c})}] =: L,
\tag{29}
$$

where $q(\boldsymbol{x}_{1:T}|\boldsymbol{x}_0, \boldsymbol{c})$ being the approximate posterior in $T$ time steps in the diffusion model (i.e., the forward diffusion process). $L$ is further decomposed into three terms by variance reduction:

$$
\begin{aligned}
L = \mathbb{E}_q[&D_{KL}(q(\boldsymbol{x}_T|\boldsymbol{x}_0, \boldsymbol{c})||p(\boldsymbol{x}_T|\boldsymbol{c})) \\
&+ \sum_{t>1} D_{KL}(q(\boldsymbol{x}_{t-1}|\boldsymbol{x}_t, \boldsymbol{x}_0, \boldsymbol{c})||p_\theta(\boldsymbol{x}_{t-1}|\boldsymbol{x}_t, \boldsymbol{c})) \\
&- \log p_\theta(\boldsymbol{x}_0|\boldsymbol{x}_1, \boldsymbol{c})].
\end{aligned}
\tag{30}
$$

In the original DDPM (Ho et al., 2020), the first term is a constant due to the fixed variance design and the last term is considered as an independent discrete decoder. Therefore, optimizing over $L$ corresponds to optimizing the second term of $L$, denoted as $L_{t-1}$. $L_{t-1}$ can be further simplified based on the assumption that all KL divergences in Eqn. 30 are comparisons between Gaussians and the posterior is tractable when conditioned on $\boldsymbol{x}_0$, which being $q(\boldsymbol{x}_{t-1}|\boldsymbol{x}_t, \boldsymbol{x}_0, \boldsymbol{c}) = \mathcal{N}(\boldsymbol{x}_{t-1}; \widetilde{\boldsymbol{\mu}}_t(\boldsymbol{x}_t, \boldsymbol{x}_0, \boldsymbol{c}), \widetilde{\beta}_t \mathbf{I})$. With specific parameterization that $p_\theta(\boldsymbol{x}_{t-1}|\boldsymbol{x}_t, \boldsymbol{x}_0, \boldsymbol{c}) = \mathcal{N}(\boldsymbol{x}_{t-1}; \boldsymbol{\mu}_t(\boldsymbol{x}_t, \boldsymbol{c}, t), \sigma_t^2 \mathbf{I})$, $L_{t-1}$ can be written as:

$$
L_{t-1} = \mathbb{E}_q[\frac{1}{2\sigma_t^2} \|\widetilde{\boldsymbol{\mu}}_t(\boldsymbol{x}_t, \boldsymbol{x}_0, \boldsymbol{c}) - \boldsymbol{\mu}_\theta(\boldsymbol{x}_t, \boldsymbol{c}, t)\|_2^2] + C,
\tag{31}
$$

where $C$ is a constant not depending on $\theta$. By reparameterization of both $\widetilde{\boldsymbol{\mu}}_t(\boldsymbol{x}_t, \boldsymbol{x}_0, \boldsymbol{c})$ and $\boldsymbol{\mu}_\theta(\boldsymbol{x}_t, \boldsymbol{c}, t)$, Eqn. 31 can be further simplified to

$$
L_{t-1} - C = \mathbb{E}_{\boldsymbol{x}_0, \epsilon}[\frac{\beta_t^2}{2\sigma_t^2 \alpha_t(1 - \bar{\alpha}_t)} \|\epsilon - \epsilon_\theta(\sqrt{\bar{\alpha}_t}\boldsymbol{x}_0 + \sqrt{1 - \bar{\alpha}_t}\epsilon, \boldsymbol{c}, t)\|_2^2],
\tag{32}
$$

where $\dfrac{\beta_t^2}{2\sigma_t^2\alpha_t(1-\bar{\alpha}_t)}$ is time step-aware fixed coefficients, $\alpha_t$ are coefficients that only relate to $\beta_t$.

As a result, minimizing Eqn. 32 corresponds to minimizing the negative log-likelihood $\mathbb{E}[-\log p_\theta(\boldsymbol{x}_0|\boldsymbol{c})]$. In practice, the simplification form:

$$\mathbb{E}_{\boldsymbol{x},\epsilon\sim\mathcal{N}(\boldsymbol{0},\boldsymbol{I}),t}[\|\epsilon - \epsilon_\theta(\boldsymbol{x}_t,\boldsymbol{c},t)\|_2^2] \tag{33}$$

is proven to be an effective and feasible approximation facilitating the training process (Ho et al., 2020). Therefore, minimizing Eqn. 33 still corresponds to minimizing the negative log-likelihood. In Sec. 2.1, following literatures, we parameterized the concept energy model $E_\psi^{concept}(\boldsymbol{x},\boldsymbol{c})$ in the form of Eqn. 33 (Eqn. 3 in ECDM), minimization of which minimizes the negative log-likelihood. The derivation above is consistent with (Ho et al., 2020), and we borrow their notation for consistency.

We also provide another perspective of Eqn. 10's simplification, the concept-based joint generation process, here:

Given the class-level instruction $\boldsymbol{y}$ and the inferred optimal concept vector $\boldsymbol{c}$, the minimization of the joint energy via sampling from the gradient of the joint energy model $\nabla_{\boldsymbol{x}}E_\psi^{joint}(\boldsymbol{x},\boldsymbol{y},\boldsymbol{c})$ can be simplified to sampling from the gradient of the concept energy network $\nabla_{\boldsymbol{x}}E_\psi^{concept}(\boldsymbol{x},\boldsymbol{c})$:

$$\nabla_{\boldsymbol{x}}E_\psi^{joint}(\boldsymbol{x},\boldsymbol{y},\boldsymbol{c}) = \nabla_{\boldsymbol{x}}E_\psi^{concept}(\boldsymbol{x},\boldsymbol{c}) \tag{34}$$

Given the instruction $\boldsymbol{y}$ and concept $\boldsymbol{c}$, we can use the Boltzmann distribution to define the conditional likelihood of the image $\boldsymbol{x}$ given $\boldsymbol{y}$ and $\boldsymbol{c}$. With the joint energy in Eqn. 7:

$$
\begin{aligned}
p(\boldsymbol{x}|\boldsymbol{c},\boldsymbol{y}) &= \frac{e^{-E_\psi^{joint}(\boldsymbol{x},\boldsymbol{c},\boldsymbol{y})}}{\sum_{\boldsymbol{x}}e^{-E_\psi^{joint}(\boldsymbol{x},\boldsymbol{c},\boldsymbol{y})}} \\
&= \frac{e^{-E_\psi^{concept}(\boldsymbol{x},\boldsymbol{c})-\lambda_m E_\psi^{map}(\boldsymbol{c},\boldsymbol{y})}}{\sum_{\boldsymbol{x}}e^{-E_\psi^{concept}(\boldsymbol{x},\boldsymbol{c})-\lambda_m E_\psi^{map}(\boldsymbol{c},\boldsymbol{y})}} \\
&= \frac{e^{-E_\psi^{concept}(\boldsymbol{x},\boldsymbol{c})}}{\sum_{\boldsymbol{x}}e^{-E_\psi^{concept}(\boldsymbol{x},\boldsymbol{c})}} = p(\boldsymbol{x}|\boldsymbol{c}).
\end{aligned}
\tag{35}
$$

Thus, we can plug Eqn. 35 into the following Bayesian formula:

$$
\begin{aligned}
p(\boldsymbol{x},\boldsymbol{c}|\boldsymbol{y}) &= p(\boldsymbol{x}|\boldsymbol{c},\boldsymbol{y}) \cdot p(\boldsymbol{c}|\boldsymbol{y}) \\
&= p(\boldsymbol{x}|\boldsymbol{c}) \cdot p(\boldsymbol{c}|\boldsymbol{y}).
\end{aligned}
\tag{36}
$$

Then take gradient with respect to $\boldsymbol{x}$ on both sides:

$$
\begin{aligned}
\nabla_{\boldsymbol{x}}\log p(\boldsymbol{x},\boldsymbol{c}|\boldsymbol{y}) &= \nabla_{\boldsymbol{x}}\log(p(\boldsymbol{x}|\boldsymbol{c}) \cdot p(\boldsymbol{c}|\boldsymbol{y})) \\
&= \nabla_{\boldsymbol{x}}\log p(\boldsymbol{x}|\boldsymbol{c}) + \nabla_{\boldsymbol{x}}\log p(\boldsymbol{c}|\boldsymbol{y}) \\
&= \nabla_{\boldsymbol{x}}\log p(\boldsymbol{x}|\boldsymbol{c}).
\end{aligned}
\tag{37}
$$

As the gradient of this energy function corresponds to the score of the conditional data distribution, we have:

$$\nabla_{\boldsymbol{x}}\log p(\boldsymbol{x},\boldsymbol{c}|\boldsymbol{y}) = \nabla_{\boldsymbol{x}}\log p(\boldsymbol{x}|\boldsymbol{c}) \iff \nabla_{\boldsymbol{x}}E_\psi^{joint}(\boldsymbol{x},\boldsymbol{y},\boldsymbol{c}) = \nabla_{\boldsymbol{x}}E_\psi^{concept}(\boldsymbol{x},\boldsymbol{c}). \tag{38}$$

## E  DATASET DETAILS

**Caltech-UCSD Birds-200-2011 (CUB).** (Wah et al., 2011) In CUB, we selected 20 classes of birds as Table 2 shows. The concept selection is identical to CBM (Koh et al., 2020). We used 60 images for each class to perform training. The class-level instruction is given as: "A photo of the bird [*bird class*]."

**Animals with Attributes 2 (AWA2).** (Xian et al., 2018) In AWA2, we selected 24 classes of animals as Table 3 shows. The concept selection is identical to ProbCBM (Kim et al., 2023). The class-level instruction is given as: "A photo of the animal [*animal class*]."

Table 2: The class selection for the CUB dataset.

| Pied billed Grebe | Purple Finch | Boat tailed Grackle | Black billed Cuckoo |
|---|---|---|---|
| European Goldfinch | Olive sided Flycatcher | Northern Fulmar | Fish Crow |
| American Crow | Scissor tailed Flycatcher | Northern Flicker | Gadwall |
| Shiny Cowbird | Eared Grebe | Great Crested Flycatcher | Vermilion Flycatcher |
| Frigatebird | Western Grebe | American Goldfinch | Horned Grebe |

Table 3: The class selection for the AWA2 dataset.

| horse | zebra | german shepherd | polar bear |
|---|---|---|---|
| sheep | rabbit | seal | grizzly bear |
| cow | lion | dolphin | giant panda |
| deer | elephant | gorilla | otter |
| squirrel | collie | buffalo | ox |
| giraffe | antelope | tiger | pig |

**CelebA-HQ.** (Karras, 2017) We selected CelebA-HQ ($1024 \times 1024$ px high resolution images), instead of CelebA ($64 \times 64$ px resolution images), to meet the demand of inputing resolution ($512 \times 512$ px) of the pretrained diffusion model. In CelebA-HQ, we performed the following procedures to curate a subset of the dataset for training: (1) Following CEM (Zarlenga et al., 2022), we screened out the top eight frequent face attributes: ['Arched Eyebrows', 'Attractive', 'Heavy Makeup', 'High Cheekbones', 'Male', 'Mouth Slightly Open', 'Smiling', 'Wearing Lipstick']. (2) We randomly selected six combinations of chosen attributes as the target class. We represented them as binaries in the Table 4. (3) We performed standard Textual Inversion (Gal et al., 2022) using the recommended default settings from Huggingface to bind each combination of concepts to an unique token (e.g., combination 1 binds to "<type1>" token). This avoided concept leakages in the training process of our model. Finally, the binded tokens were used as the class-level instructions in our model. The class-level instruction is given as: "A photo of the face [*unique token*]."

Table 4: The token-attribute relationship in CelebA-HQ dataset.

| Attributes / Tokens | Arched Eyebrows | Attractive | Heavy Makeup | High Cheekbones | Male | Mouth Slightly Open | Smiling | Wearing Lipstick |
|---|---|---|---|---|---|---|---|---|
| <type1> | 1 | 1 | 1 | 0 | 0 | 1 | 0 | 1 |
| <type2> | 0 | 0 | 0 | 1 | 1 | 1 | 1 | 0 |
| <type3> | 0 | 1 | 0 | 1 | 0 | 1 | 1 | 1 |
| <type4> | 1 | 0 | 0 | 1 | 1 | 1 | 1 | 0 |
| <type5> | 1 | 1 | 1 | 0 | 0 | 0 | 0 | 1 |
| <type6> | 1 | 1 | 0 | 1 | 0 | 1 | 1 | 1 |

## F IMPLEMENTATION DETAILS

**Association Between Model Parameters and Concept Number.** We further provide the scaling association between the model parameters and the concept number as Fig. 6 shows.

**Sampling Efficiency.** We sample from the mapping energy network using Gradient Inference technique, following (Xu et al., 2024). $10 \sim 30$ steps, corresponding to approximately 10 seconds wall-clock time, is needed for this sampling procedure. For the generative concept energy network, we model the diffusion model as an implicit modeling of the energy function. Therefore, the diffusion model sampling algorithm is applicable to our framework. We leveraged standard diffusion sampling algorithm (DDIM (Song et al., 2020a)) to generate an image from the concept energy network. 50 steps, corresponding to approximately 3 seconds wall-clock time when using Nvidia RTX 3090, is needed for this sampling procedure.

**Training Details.** We build our model based on publicly available Stable Diffusion 2.1 model, and $512 \times 512$ as the input size for all evaluated methods, unless stated otherwise. We use the AdamW optimizer to train the model. We use $\lambda_m = 0.1$, batch size 4, a learning rate of $4 \times 10^{-3}$, and at most 100 iteration per image. We run all experiments on two NVIDIA RTX3090 GPUs. To perform

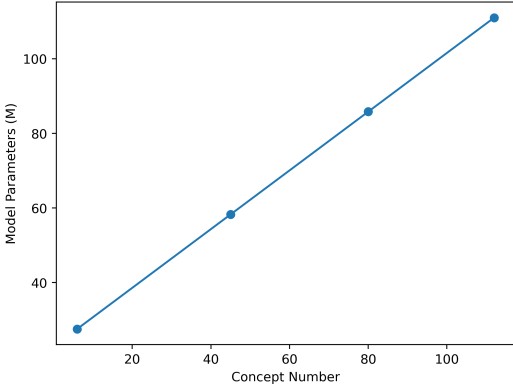

Figure 6: Association Between Model Parameters and Concept Number. The energy estimation network and feature mappings of our proposed method is efficient, and scales linearly with the number of concepts. We exclude all frozen pretrained parameters in the parameter counting.

negative sampling in the training process, we perturb $30\%$ of the concept set to sample 2 negative concept vectors per positive sample. These are not incorporated in the generation and interpretation process.

**Generation Details.** For all trained diffusion models, we use the same generation sampler (DDIM Sampler), sampling steps (50 steps), and random seed as recommended by Huggingface. All class-level instructions are consistent per dataset among all methods.

### F.1 DETAILS OF THE CONCEPT-INVERSION INTERPRETATION

**DDIM Inversion.** Given an image $x_0$, DDIM sampling (Song et al., 2020a) provides a path that allows inverting the image back to the noised latents based on the assumption that ODE can be inverted in the limit of sufficiently small steps (Kim et al., 2021). The inversion path is:

$$x_{t+1} = \sqrt{\frac{\alpha_{t+1}}{\alpha_t}}x_t + \left(\sqrt{\frac{1}{\alpha_{t+1}} - 1} - \sqrt{\frac{1}{\alpha_t} - 1}\right) \cdot \epsilon_\theta\left(y, x_t, t\right), \qquad (39)$$

where $\alpha_t$ is the noise scheduling coefficient at timestep $t$ provided by the DDIM scheduler. This inversion path enables a replay of the sampling trajectory, hence facilitating meaningful editing (Kim et al., 2021; Mokady et al., 2023) or interpretation. Similar to Eqn. 16~18, one can replace $y$ with $c$. We built our Concept Inversion based on the reverse DDIM detailed as follows.

According to Classifier-Free Guidance (Ho & Salimans, 2022), we can obtain a better conditional diffusion model output $\epsilon_\theta\left(y, x_t, t\right)$ to be used in the Eqn. 39 by performing:

$$\widetilde{\epsilon}_{\boldsymbol{\theta}}(y, x_t, t) = \epsilon_{\boldsymbol{\theta}}(\varnothing, x_t, t) + w(\epsilon_{\boldsymbol{\theta}}(y, x_t, t) - \epsilon_{\boldsymbol{\theta}}(\varnothing, x_t, t)), \qquad (40)$$

where $\epsilon_{\boldsymbol{\theta}}(\varnothing, x_t, t)$ denotes unconditional diffusion model (giving the model input unconditional embedding in implementation), and $w$ can be seen as the conditional guidance strength. We adopt this guidance strategy in the sampling process of ECDM to obtain conditional diffusion model's final outputs. Several studies (Mokady et al., 2023; Dong et al., 2023) have found that the selection of guidance strength $w$ have strong effect in the reverse DDIM process: lower $w$ (e.g., $w = 1$) increases the fidelity of the recovered image based on the reverted path, while higher $w$ (e.g., $w = 7.5$) ensures a better edit ability based on the reversed path. The complication of higher $w$ is the increase of ODE sampling error, making the generated sample deviate from the reversed trajectory. To make the best of both worlds, we used a three stepped strategy to (1) retain the original conditional sampling trajectory for interpretation (energy matching), (2) enable the intervention ability based on the interpreted trajectory by using higher $w$, and (3) cancel out the deviating error brought by the larger value of $w$.

**Step 1: Pivotal Inversion.** In the inversion process, we reverse a generated image $x$ back to a trajectory of noised latent by using Eqn. 39 and $w = 1$. By using $w = 1$ the diffusion model would only output the instruction-conditioned output $\epsilon_\theta\left(y, x_t, t\right)$, hence a better depiction of the

distribution $p(\boldsymbol{x}|\boldsymbol{y})$ for the subsequent matching process. The reversed trajectory is saved for the following process.

**Step 2: Error Cancellation by Null Text Optimization.** We used the reversed trajectory saved in step 1 to perform the replay of the generation sampling process. Inspired by (Mokady et al., 2023), we adopted the same strategy in this step. We use larger $w = 7.5$ to obtain a conditional diffusion model output for better edibility, and optimize the unconditional embedding per sampling step $\bar{\varnothing}_t|_{t=1,\dots,T}$ to cancel out the sampling error. The optimized unconditional embeddings are saved for the use of step 3.

**Step 3: Concept Inversion by Incorporating the Optimized Unconditional Embedding.** In this step, we start again from the noised latent to perform generation sampling prediction but incorporating $\bar{\varnothing}_t|_{t=1,\dots,T}$ and use $w = 7.5$. Specifically, we generate the output to perform matching by:

$$\widetilde{\epsilon}_{\boldsymbol{\theta}}(\boldsymbol{c}, \boldsymbol{x}_t, t) = \epsilon_{\boldsymbol{\theta}}(\bar{\varnothing}_t, \boldsymbol{x}_t, t) + w(\epsilon_{\boldsymbol{\theta}}(\boldsymbol{c}, \boldsymbol{x}_t, t) - \epsilon_{\boldsymbol{\theta}}(\bar{\varnothing}_t, \boldsymbol{x}_t, t)), \tag{41}$$

where $\boldsymbol{c}$ is the concept vector, the only vector we optimize in this step to obtain the concept probability.

By this means, both the edibility and the interpretability are preserved in the Concept Inversion process. In practice, the second step is efficient with the early stopping strategy proposed in (Mokady et al., 2023).

## F.2 EVALUATION DETAILS

**Evaluation Sample Number.** To match the amount of the reference image when calculating FID, we used 2400, 1200, and 600 synthetic images for AWA2, CUB, and CelebA-HQ dataset, respectively. All methods generated the same amount of images for evaluation.

**Details of the Classifier Used for Class Accuracy Calculation.** We used ResNet101 (He et al., 2016) to train classifiers on real images of these dataset to assess class accuracy. We used the official data splits and recommended default hyperparameters for classifier training. The accuracy of these three classifiers on CUB, AWA2, and CelebA-HQ real image test sets are: 0.7561, 0.9230, and 0.9526.

**Details of the Classifier Used for Concept Accuracy Calculation.** We used CEM (Zarlenga et al., 2022) to train concept prediction models on real images of these dataset to assess concept accuracy. CEM employed individual concept classifiers to predict each concept, achieving higher task performance than the vanilla CBM (Koh et al., 2020) while maintaining high prediction efficiency, hence become the choice. We used the official data splits and recommended default hyperparameters in the official implementation for classifier training. The performance of these three CEM classifiers on CUB, AWA2, and CelebA-HQ real image test sets are: 0.9649, 0.9810, and 0.9042.

**Reproducibility.** We will release the code upon the publication of this paper.

