# OpenReview forum: "Energy-Based Conceptual Diffusion Model"
_NeurIPS.cc/2024/Workshop/SafeGenAi — SafeGenAi Poster_

### Official Review · Reviewer_duMx · 2024-10-08
**Paper Review: "Energy-Based Conceptual Diffusion Model"**

**Rating:** 10
**Confidence:** 4

**Review:**

Innovative Integration: The paper introduces an innovative integration of Energy-Based Models (EBMs) with diffusion models and Concept Bottleneck Models (CBMs), enhancing the interpretability and control in generative tasks. This integration allows for a more structured and interpretable generation process compared to traditional diffusion models.
Enhanced Control and Debugging: The proposed model, ECDM, enables detailed control over the generation process through concept-based interventions and corrections. This is particularly useful for applications requiring high fidelity and precision, such as medical imaging or detailed artistic creations.
Systematic Interpretation: ECDM provides a systematic framework for interpreting the generation process, which is a significant advancement over existing models that offer limited interpretability. This feature is crucial for understanding model behavior and ensuring reliability in automated systems.
Empirical Validation: The paper presents comprehensive experiments across various datasets, demonstrating the model's effectiveness in generating high-quality images with accurate concept representation. The empirical results underline the model's practical utility and robustness.
Concept-Based Interventions: The ability to perform concept-based corrections and imputations based on the identified errors during the generation process is a significant advantage, offering potential for practical applications where precision is critical.

---

### Official Review · Reviewer_RtE3 · 2024-10-09
**Good paper, accept**

**Rating:** 7
**Confidence:** 3

**Review:**

### Summary
The authors tackle the problem of limited concept accuracy in diffusion models and propose a way to unify concept, label, and image generation using energy based models conditioned on label and concept, respectively. Subsequently, they go on to provide experiment results on AWA2, CUB, and CelebA-HQ, showing their method's effectiveness.

### Pros
- Clear writing
- The proposed method is simple

### Cons/Questions
- How do sample from the energy based models in a cheap way? I believe there needs to be a wall-clock time evaluation.
- How does the method scale with increasing values of $K$?